# Study on Cellulose Acetate Butyrate/Plasticizer Systems by Molecular Dynamics Simulation and Experimental Characterization

**DOI:** 10.3390/polym12061272

**Published:** 2020-06-02

**Authors:** Weizhe Wang, Lijie Li, Shaohua Jin, Yalun Wang, Guanchao Lan, Yu Chen

**Affiliations:** School of Material Science and Engineering, Beijing Institute of Technology, Beijing 100081, China; waxwellwang@163.com (W.W.); lilijie2003@bit.edu.cn (L.L.); jinshaohua@bit.edu.cn (S.J.); 3120181161@bit.edu.cn (Y.W.); lan890805@163.com (G.L.)

**Keywords:** cellulose acetate butyrate, plasticizer, molecular dynamics simulation, thermal analysis, compatibility

## Abstract

Cellulose acetate butyrate (CAB) is a widely used binder in polymer bonded explosives (PBXs). However, the mechanical properties of PBXs bonded with CAB are usually very poor, which makes the charge edges prone to crack. In the current study, seven plasticizers, including bis (2,2-dinitro propyl) formal/acetal (BDNPF/A or A3, which is 1:1 mixture of the two components), azide-terminated glycidyl azide (GAPA), n-butyl-N-(2-nitroxy-ethyl) nitramine (Bu-NENA), ethylene glycol bis(azidoacetate) (EGBAA), diethylene glycol bis(azidoacetate) (DEGBAA), trimethylol nitromethane tris (azidoacetate) (TMNTA) and pentaerythritol tetrakis (azidoacetate) [PETKAA], were studied for the plasticization of CAB. Molecular dynamics simulation was conducted to distinguish the compatibilities between CAB and plasticizers and to predict the mechanical properties of CAB/plasticizer systems. Considering the solubility parameters, binding energies and intermolecular radical distribution functions of these CAB/plasticizer systems comprehensively, we found A3, Bu-NENA, DEGBAA and GAPA are compatible with CAB. The elastic moduli of CAB/plasticizer systems follow the order of CAB/Bu-NENA>CAB/A3>CAB/DEGBAA>CAB/GAPA, and their processing property is in the order of CAB/Bu-NENA>CAB/GAPA>CAB/A3>CAB/DEGBAA. Afterwards, all the systems were characterized by FT-IR, differential scanning calorimetry (DSC), differential thermogravimetric analysis (DTA) and tensile tests. The results suggest A3, GAPA and Bu-NENA are compatible with CAB. The tensile strengths and Young’s moduli of these systems are in the order of CAB/A3>CAB/Bu-NENA>CAB/GAPA, while the strain at break of CAB/Bu-NENA is best, which are consistent with simulation results. Based on these results, it can be concluded that A3, Bu-NENA and GAPA are the most suitable plasticizers for CAB binder in improving mechanical and processing properties. Our work has provided a crucial guidance for the formulation design of PBXs with CAB binder.

## 1. Introduction

Cellulose acetate butyrate (CAB), with the molecular formula of [C_6_H_7_O_2_-(OCOCH_3_)_X_-(OCOC_3_H_7_)_Y_-(OH)_3-X-Y_]_n_, is the most commonly used binder in polymer bonded explosives (PBXs). In the structure of cellulose ester, hydroxyl groups are co-esterified with acetic acid and butyric acid [1]. Therefore, CAB contains about 12–15% (wt %) of acetyl groups and 26–39% (wt %) of butyryl groups, which endows it with excellent physical and chemical properties including outstanding moisture resistance, UV resistance, temperature resistance, flexibility, transparency, etc. Therefore, many researchers have utilized this excellent cellulose derivative in the field of energetic materials. Li prepared the CL-20 based pressed PBXs with CAB as binder for its excellent compatibility and mechanical properties [2]. Lan ameliorated the HNIW based PBXs with excellent temperature adaptability using CAB and fluorine rubber F2311 as binders [3].

However, the CAB-bonded explosives exhibit some defects and encounter problems in their practical applications. The softening temperature (T_S_) and glass transition temperature (T_g_) of CAB are relatively high. Therefore, its molecular chain cannot stretch sufficiently and its adhesion is weak in solvents. The PBXs with CAB as binder usually exhibit poor mechanical properties at low temperatures, unsatisfactory insensitivity, and deterioration of cracks [4].

To ameliorate these situations, plasticizers have been added to the CAB binder system to lower its T_S_ and T_g_, increase its plasticity, improve the mechanical properties and insensitivity, and prevent the deterioration of cracks on PBXs [5]. However, the effects of different plasticizers on properties of PBXs with CAB binder remain unclear. Therefore, it is essential to select optimal plasticizers for CAB binder system of PBXs. Experimental tests are time-consuming and inconvenient for repeated comparisons, accompanied by certain risks. In addition, it cannot be used for the in-depth analysis from the microscopic perspective.

Molecular dynamics (MD) simulation has been widely used to predict the compatibility and mechanical properties of blends [6,7,8,9]. Compared with the macroscopic experiments, MD simulations in microscopic scales not only can reveal the effects of plasticizer at the atomic and molecular levels, but also is low-cost and time efficient. Therefore, it can be used to study the effects of different plasticizers on CAB binder quickly and efficiently [10]. In the present work, the compatibilities between CAB and different plasticizers were evaluated by MD simulation. The mechanical properties of the plasticized CAB with selected plasticizers were then simulated. Three representative CAB/plasticizer systems were prepared, and their compatibilities, chemical stability and mechanical properties were characterized experimentally, aiming to establish a study method combining MD simulation and experimental characterization for optimizing CAB/plasticizer systems. Our work has provided a guidance and reference for the subsequent modification of CAB binder and the formulation design of PBXs using CAB binder.

## 2. MD Simulation

Based on their molecular structures (Figure 1), the molecular models of CAB and plasticizers including bis (2,2-dinitro propyl) formal/acetal (BDNPF/A or A3, which is 1:1 mixture of the two components), azide-terminated glycidyl azide (GAPA), n-butyl-N-(2-nitroxy-ethyl) nitramine (Bu-NENA), ethylene glycol bis(azidoacetate) (EGBAA), diethylene glycol bis(azidoacetate) (DEGBAA), trimethylol nitromethane tris (azidoacetate) (TMNTA) and pentaerythritol tetrakis(azidoacetate) (PETKAA) were constructed using the Visualizer module in Material Studio (MS) version 6.0. The butyryl group and acetyl group contents in CAB are approximately set to 37 wt % and 13 wt %, respectively. Anneal and geometric optimizations were then conducted to establish the final molecular models.

### 2.1. Construction of Models

To evaluate the compatibility between CAB and plasticizers and predict the mechanical properties of the plasticized CAB, MD simulation was conducted in the Material Studio with use of COMPASS force field [11].

The next simulation section is the annealing of various molecular models, which aims to relax their configurations, lower the potential energy and eliminate the internal stress of polymer chains. The initial temperature and mid-cycle temperature were set to 298K and 500K, respectively. The temperature range covers different glass transition temperatures (T_g_) of CAB and plasticizers (CAB: 403.2K, BDNPF/A: 208.6K, Bu-NENA: 189.7K, EGBAA: 203.7K, DEGBAA: 209.9K, GAPA: 194.4K, PETKAA: 237.8K, TMNTA: 239.1K) [12,13,14,15,16,17,18]. The heating ramps per cycle was 25, which was a proper step number. Too rapid cooling often traps the system in a high energy–low density state that cannot represent the glassy state. In the contrast, slow annealing from rubbery state with high T_g_ can generate more accurate results. After 500 cycles of annealing, the annealed molecular models were obtained for further simulation.

The amorphous models of CAB/plasticizer systems were then established using the Amorphous Cell module as shown in Figure 1. For the construction of these amorphous models, the quality was set to fine, the maximum number of steps was set to 50,000, and the density of the mixed system was calculated based on the mass ratio of CAB/plasticizer. Ewald was set as the electrostatic force, van der Waals was set as the atom based, and cutoff radius was set to 12.50 Å for the simulation. The mass ratio of CAB/plasticizer was set to 2:3, which is a commonly used ratio in PBXs, and the amounts of CAB molecules (3 chains, 734 atoms) in all amorphous models were the same. The plasticizer contents were approximately in the range of 30–60 molecules (1200–1800 atoms) in different systems, depending on the plasticizer.

### 2.2. Calculations

The geometric structures of CAB/plasticizer amorphous models were optimized by “Geometry Optimization” option in the Forcite module. The MD simulation of 250 ps was performed in NVT ensemble at the temperature of 298K to achieve the optimal amorphous models [19]. COMPASS force field was applied with Andersen thermostat for the simulation. Figure 2 shows the dynamics temperature-time curve and energy-time curves obtained by MD simulation. The simulation is considered to reach the equilibrium state if the fluctuations of dynamic temperature and dynamic energy are less than 5% [20]. The dynamic energy may seem almost constant during the simulation, but it changes slightly with time, which is so-called “aging”.

After dynamics simulation reaching the equilibrium state, the calculated dynamic trajectory files were analyzed by the “Cohesive Energy Density” option of the Forcite module to obtain the cohesive energy density and solubility parameters of CAB and plasticizers.

The binding energies of CAB, plasticizers and their mixed systems were produced by the analysis with “Total Kinetic Energy” option in Forcite module. Five frames of each model with lowest energy were selected and their binding energy parameters were obtained with “Energy” option in Forcite, which were then averaged.

Radial distribution functions (RDF) were obtained by the analysis with “Radial distribution function” option of Forcite module. The dynamic trajectory files of the CAB molecules and plasticizers were classified to different sets and analyzed with “Radial distribution function” option to give RDFs.

The mechanical properties of the systems were simulated in a more complex manner. The amorphous models of CAB/plasticizers were firstly subjected to the 250 ps dynamics simulation with NPT ensemble under 1GPa. The last frame model of previous dynamic trajectory files was extracted to subject to another 250 ps dynamics simulation with the NPT ensemble under 1atm. The last frame model was then extracted to undergo the 250 ps dynamics simulation with NVT ensemble and the final dynamic trajectory files were calculated by “Mechanical Properties” option in Forcite module. The simulation tests were performed by the “Constant strain” method. The number of steps for each strain was set to 4 and the maximum strain was 3 × 10^−3^. The strain rate was used as the automatic systematical setting.

## 3. Experiments and Characterization

### 3.1. Materials

Three representative plasticizers, A3, GAPA and Bu-NENA, were selected for experimental characterization. Analytical grade CAB containing 37 wt % butyryl group and 13 wt % acetyl group was purchased from Eastman Chemical Company (USA). A3 and Bu-NENA (purity > 99.9%) were provided by Liming Chemical Research Institute (Henan, China). GAPA (purity > 99.12%, ρ = 1.21 g/cm^3^, water content < 0.016%, hydroxyl value ≈ 1.5 mg KOH/g, molecular weight Mn ≈ 800 g/mol) was synthesized in-house.

### 3.2. Preparation of CAB/Plasticizer Mixed Systems

CAB and plasticizer were mixed at the mass ratio of 2:3. For a typical procedure, 2.00 g CAB was dissolved in 20 mL ethyl acetate in a 50 mL flask by magnetic stirring, and 3.00 g plasticizer was then added into the flask and mixed well with CAB solution. Ethyl acetate was then removed by evaporation under reduced pressure with a rotary evaporator to afford the CAB/plasticizer mixture.

### 3.3. Characterizations

The structure of CAB/plasticizer was characterized using a Fourier transform infrared spectroscopy (FT-IR) analyzer (NEXUS-470, Nicolet, WI, USA) by the KBr pellet method.

The compatibilities between CAB and plasticizers were respectively evaluated by differential scanning calorimetry (DSC) using a DSC131 Evo instrument (Setaram, France) and differential thermogravimetric analysis (DTA) using a DTA-60 thermal analyzer (Shimadzu, Japan) under 50 mL/min nitrogen flow atmosphere at the heating rate of 10 K/min.

The tensile strengths of the mixed systems were measured using a universal material testing machine (6022, Instron, Norwood, MA, USA) at the loading rate of 100 mm/min until broken. Ten test specimens were prepared and characterized, whose size were 20 mm long and 5 mm wide.

## 4. Results and Discussion

### 4.1. MD Simulation

#### 4.1.1. Solubility Parameter

According to the polymer solution theory, the mixture of polymer and plasticizer can be considered as polymer solution system [21]. The necessary thermodynamic condition for a spontaneous dissolution at constant temperature under constant pressure is:(1)$$\Delta G_{M}=\Delta H_{M}-T\Delta S_{M}<0$$
where Δ*G* is the free energy of mixing, Δ*S* is the entropy of mixing, Δ*H* is the heat of mixing, and *T* is the temperature of dissolution. As a polymer is mixed with a plasticizer, the polymer itself is in a chaotic state. Therefore, the magnitude of Δ*G* mainly depends on Δ*H* [22]. Hildebrand introduced the concept of solubility parameter (*δ*) [23], which was defined as the square root of the cohesive energy density (CED) (Equation (2)).
(2)$$\delta ={\left(\Delta E/V\right)}^{1/2}={\left[\left(\Delta H-RT\right)/V\right]}^{1/2}$$

The heat of mixing Δ*H_M_* of the polymer dissolution can be expressed using the Hildebrand formula of small molecules:(3)$$\Delta H_{M}=V_{M}\Phi_{1}\Phi_{2}{\left[{\left(\Delta E_{1}/V_{1}\right)}^{1/2}-{\left(\Delta E_{2}/V_{2}\right)}^{1/2}\right]}^{2}$$
where Φ_1_ and Φ_2_ are the volume fractions of the two components, respectively, and *V* is the molar volume of the mixture. If the square root of CED is replaced with *δ*, Equation (3) can be re-written as:(4)$$\Delta H_{M}=V_{M}\Phi_{1}\Phi_{2}{\left(\delta_{1}-\delta_{2}\right)}^{2}$$

Therefore, the value of Δ*H_M_* is determined by *δ*_1_ and *δ*_2_. The closer *δ*_1_ and *δ*_2_ are, the smaller Δ*H_M_* is and the better the compatibility of the two components will be. For an energetic material, its polymer and plasticizer are generally considered compatible if |Δ*δ*| < 3.68~4.06 (J^1/2^·cm^−3/2^) [24].

Table 1 lists the solubility parameters (*δ_MD_*) of CAB and each plasticizer, and the respective differences between the solubility parameters of CAB and those of different plasticizers (|Δ*δ_MD_*|) obtained by the MD simulation. The difference between the simulation *δ* of CAB (17.29) and reported *δ* (18.87) is very small [25,26], indicating that although MD simulation produces statistical uncertainties from the ideal simulation conditions and imperfection algorithm, the simulation results are mostly consistent with reality.

The |Δ*δ_MD_*| values between CAB and A3, GAPA, DEGBAA and Bu-NENA plasticizer are less than 3.55, suggesting that these plasticizers are compatible with CAB. However, the values of |Δ*δ_MD_*| between CAB and EGBAA, TMNTA and PETKAA are greater than 6.00, indicating that these plasticizers are unsuitable for the plasticization of CAB.

#### 4.1.2. Binding Energy

The good compatibility between a polymer and a different compound implies strong intermolecular interaction(s) between them. The strength of such interaction can be quantitatively described with binding energy (*E_bind_’*). The average interaction energy between CAB and a plasticizer (*E_inter_*) can be defined as:(5)$$E_{inter}=E_{total}-\left(E_{CAB}+E_{plasticizer}\right)$$
where *E_total_* is the total energy of the mixed system, and *E_CAB_* and *E_plasticizer_* are the average energies of CAB and plasticizer, respectively. The binding energy *E_bind_* is the negative value of the average interaction energy *E_inter_*, e.g.:(6)$$E_{bind}=-E_{inter}$$

Because the average molecular weights of the mixed systems are different, the binding energy is converted into per unit mass *E_bind_’* for comparison purpose as shown in Equation (7):(7)$$E_{bind}{}^{\prime }=E_{bind}/M_{n}$$
where *M_n_* is the average molecular weight calculated based on the mass ratio of CAB/plasticizer.

Table 2 lists the *E_bind_* and *E_bind_’* of each CAB/plasticizer system. No chemical bonds contribute to the binding energy. The binding energies are numerically equal to the corresponding non-bonding energies and follow the order of CAB/Bu-NENA≈CAB/EGBAA> CAB/DEGBAA> CAB/TMNTA> CAB/PETKAA> CAB/GAPA> CAB/A3. It is known that the system with greater *E_bind_’* has better compatibility [27]. Therefore, CAB/A3 is the least compatible system and all other CAB/plasticizer systems are more compatible. The discrepancy with Solubility Parameter simulation results is mainly generated from the ideal simulation conditions and imperfection algorithms. Although the solubility parameter is the most reliable criteria for compatibility, binding energy can also provide supplementary information.

#### 4.1.3. Radial Distribution Function (RDF)

In statistical mechanics, the RDF of a system of different particles (atoms, ions, molecules, etc.) is applicable to the structural investigations of both solid and liquid packing (local structure) for studying specific interactions, such as hydrogen bonding. It measures the probability of finding a target particle at the distance of *r* around the given reference particle [28]. In other words, it describes how the density varies with distance from a reference particle [29]. In a blend of various molecules, two different components tend to be compatible with each other if the RDF curve of mixed system is higher than each of their own [30]. Figure 3 shows the radial distribution functions (RDFs) of different CAB/plasticizer systems and those of their individual components.

The RDFs of CAB and all plasticizers reach the peak values at the distances slightly shorter than 5.00–6.00 Å. It is well known that the distance ranges of hydrogen bond and van der Waals force are 0–3.10 Å and 3.10–5.00 Å, respectively [31,32,33,34]. Therefore, van der Waals force is the major intermolecular forces of CAB-CAB and plasticizer-plasticizer pairs. The RDF of CAB/plasticizer is lower than that of plasticizer, but higher than CAB itself in the range of 0–6.00 Å for all systems, suggesting that the compatibility between CAB and plasticizers seems not so good.

The RDF curves of CAB/Bu-NENA, CAB/GAPA and CAB/PETKAA are higher than themselves at the distances longer than 12.47 Å, 12.94 Å and 14.09 Å, which are far beyond the range of intermolecular forces, indicating these plasticizers are not compatible with CAB. The nitro and azide groups in their structure may be the main reason of the weak physical interactions between these groups and cellulose chains of CAB.

The RDF curves of CAB/A3, CAB/DEGBAA, and CAB/TMNTA are higher than those of the corresponding plasticizers at distances longer than 10.71 Å, 8.41 Å and 10.17 Å, indicating that these plasticizers are compatible with CAB. These results suggest that compatibilities of CAB/A3, CAB/DEGBAA, and CAB/TMNTA are better than those of CAB/GAPA, CAB/Bu-NENA and CAB/PETKAA.

Considering three compatibility criteria including solubility parameter, binding energy and radial distribution function (RDF) comprehensively, it can be concluded that GAPA, Bu-NENA, A3 and DEGBAA are compatible with CAB, and PETKAA, TMNTA and EGBAA are incompatible with CAB.

#### 4.1.4. Simulation of Mechanical Properties

The mechanical properties of an energetic material greatly affect the safety and storage performance of its explosive products, which thus are of significant importance [35]. Bulk modulus (*K*), shear modulus (*G*), Poisson’s ratio (*μ*), Young’s modulus (*E*), etc. are usually used to describe the mechanical properties of energetic materials.

The volume of a material decreases under uniform pressure *P*. Therefore, bulk modulus is defined as:(8)$$K=\frac{P}{-\Delta V/V}=-\frac{PV}{\Delta V}$$
where *V* is the volume of material and Δ*V* is the volume change under pressure *P*. Bulk modulus is a measure of compressibility and breaking strength of a material. The larger the bulk modulus, the higher the breaking strength [36].

Shear modulus (*G*) is the ratio of shear stress (*σ*) to shear strain (*γ*):(9)$$G=\frac{\sigma }{\gamma }=\frac{P/A}{tg\theta }$$

It is a measure of the stiffness of a material. The greater the value of shear modulus, the higher the material hardness and the smaller the deformation.

The ratio of bulk modulus to shear modulus (*K/G*) reflects the extent of plastic change (elongation in tension) of the material. The greater the value of *K/G* is, the better the ductility of the material is [37].

Young’s modulus (*E*) is defined as the ratio of tensile stress (σ) to tensile strain (*ε_1_*):(10)$$E=\frac{\sigma }{\varepsilon_{1}}$$

It can be used to evaluate the capability of a material to resist deformation and volume change caused by external stresses.

Poisson’s ratio (μ) is defined as the ratio of transverse shrinkage deformation (*ε_2_*) to longitudinal stretch deformation (*ε_1_*):(11)$$\mu =-\frac{\varepsilon_{2}}{\varepsilon_{1}}$$

It is an elastic constant reflecting the transverse deformation of a material. In general, the *μ* less than 0.5 under tensile stress results in volume increases. The materials with the Poisson’s ratios in the range between 0.2 and 0.4 are generally considered to have good plasticity [38,39,40,41].

The moduli of a typical isotropic material satisfy the following relationship [42]:(12)E=2G(1+μ)=3K(1−2μ)

Table 3 lists the bulk moduli (*K*), shear moduli (*G*) and other calculated mechanical parameters of the CAB/plasticizer systems obtained by MD simulation. CAB/Bu-NENA exhibits the largest bulk modulus (*K*), indicates that its breaking strength is the highest. The shear moduli (*G*) of the CAB/plasticizer systems are similar, within the range of 0.8–0.9 GPa. The Poisson’s ratio (*μ*) and the *K/G* value of CAB/Bu-NENA are also the greatest, indicating its ductility and formation property are the best. The Young’s modulus (*E*) of the CAB/Bu-NENA system is greatest, suggesting its resistance to deformation is the highest. The discrepancy between simulation results and actual mechanical parameters is mainly generated from the ideal simulation conditions and imperfection algorithms.

Based on the simulated mechanical properties obtained above, it can be concluded that the mechanical properties of CAB/Bu-NENA are the best, followed by those of CAB/A3 and CAB/GAPA. The mechanical properties of CAB/DEGBAA are poorest.

### 4.2. Experimental Characterization

Based on the simulation results, CAB/A3, CAB/GAPA and CAB/Bu-NENA mixed systems were prepared, and their compatibilities and mechanical properties were characterized by FT-IR, DSC, DTA and tensile strength tests for comparison purpose. The feasibility to optimize the plasticity of binder and predict the mechanical properties of the plasticized binder by combining theoretical simulation and experimental characterization was further demonstrated.

#### 4.2.1. Chemical Stability

The chemical stability of an energetic material can be determined by FT-IR. If the characteristic peaks of mixture are the same as those of raw materials, it can be considered no chemical change occurs during the mixing [43]. Figure 4 shows the FT-IR spectra of CAB, plasticizers and their mixed products.

The CAB/GAPA system is analyzed as an example. CAB exhibits a broad and strong absorption peak at 3445.86 cm^−1^ due to the stretching vibration of -OH. The doublet peak at 2964.33 cm^−1^ and 2877.88 cm^−1^ are attributed to the antisymmetric and symmetric stretching vibrations of methylene group. The strong absorption peak at 1742.35 cm^−1^ can be assigned to the stretching vibration of -C=O. The peaks at 1165.11 cm^−1^ and 1065.15 cm^−1^ are ascribed to the stretching vibration of the unique -COCOC-polyether structure in CAB.

GAPA shows a doublet peak at 2929.89 cm^−1^ and 2877.42 cm^−1^ that can be assigned to the antisymmetric and symmetric stretching vibration of methylene. The strong peak at 2100.68 cm^−1^ and the peak at 1281.22 cm^−1^ are the characteristic absorption peaks of GAPA caused by the stretching vibration and bending vibration of -N_3_, respectively. The peak at 1128.16 cm^−1^ can be assigned to the antisymmetric stretching vibration of ether bond.

The stretching vibration of -OH remains at 3462.72 cm^−1^ in the CAB/GAPA mixture. The peaks of the mixture at 2934.93 cm^−1^ and 2879.95 cm^−1^ are due to the superposition of the antisymmetric and symmetric stretching vibration absorption peaks of the methylene groups in CAB and GAPA. The stretching vibration peak of -N_3_ of GAPA shifts to 2097.35 cm^−1^. The stretching vibration peak of -C=O of CAB is found at 1730.42 cm^−1^. The peak at 1275.13 cm^−1^ is attributed to the bending vibration of the -N_3_ of GAPA. The stretching vibration of -COCOC- polyether structure of CAB results in the absorption peak at 1202.30 cm^−1^. Based on these results, it can be concluded that no chemical reaction occurs during the mixing, and thus the chemical properties of the mixture systems are relatively stable.

Similar results are obtained for the CAB/Bu-NENA and CAB/A3 systems. Therefore, mixing CAB with plasticizers does not change the chemical properties of the individual components, and the CAB/plasticizer systems are chemically stable.

#### 4.2.2. Compatibility

The compatibilities between an explosive and its contacting materials can be evaluated by DCS and DTA. According to the National Military Standard of China GJB 772A-97 502.1, the compatibility can be classified into four levels: level A with Δ*T_p_* ≤ 2.0/°C and Δ*E/E_a_* ≤ 20%, the system is compatible or highly compatible; level B with Δ*T_p_* ≤ 2.0/°C and Δ*E/E_a_* > 20%, the system is slightly sensitized or fairly compatible; level C with Δ*T_p_* > 2.0/°C and Δ*E/E_a_* ≤ 20%, the system is sensitized or poorly compatible; level D with Δ*T_p_* > 2.0/°C and Δ*E/E_a_* > 20%, the system is hazardous. In the standard, Δ*T_p_* is the change of the decomposition exothermal temperature, and Δ*E/E_a_* is the changing rate of the apparent activation energy.

Figure 5 shows the DSC and DTA curves of CAB, the plasticizers and the CAB/plasticizer systems at the heating rate of 10 K/min. Table 4 lists the compatibilities of CAB with three energetic plasticizers obtained by DSC and DTA.

The first exothermic peak of A3 appears at 263.66 °C, which is caused by the main decomposition. The secondary decomposition of the partial decomposition products appeared at 432.53 °C. The main exothermic peak of CAB/A3 is found at 265.23 °C. The peak temperature shift Δ*T_p_* is within 2 °C and the Δ*E/E_a_* is calculated to be 17.25%, indicating the compatibility of CAB/A3 system reaches level A. The exothermic peak temperature of Bu-NENA is 213.91 °C, and the main exothermic peak of CAB/Bu-NENA is at 215.79 °C. The peak temperature shift Δ*T_p_* is within 2 °C and the calculated Δ*E/E_a_* is 16.62%, suggesting the compatibility of CAB/Bu-NENA system is also level A. Similarly, the compatibility of CAB/GAPA system is found to be level A with the exothermic peak temperatures of GAPA and CAB/Bu-NENA respectively at 252.02 and 253.63 °C, and the Δ*E/E_a_* of 18.05%.

Similar results are also obtained by DTA (Table 4). The peak temperature shifts of all three CAB/plasticizer systems are within 2 °C and the corresponding Δ*E/E_a_* values are less than 20.00%. These results suggest that A3, Bu-NENA and GAPA are compatible with CAB at level A, and thus can be safely used in explosive design. In addition, the experimental compatibility results obtained by DSC/DTA are consistent with MD simulation results, indicating MD simulation is applicable to the characterization of the plasticized binders for the design of energetic materials.

#### 4.2.3. Mechanical Properties

Figure 6 shows the tensile strengths (*σ*), Young’s moduli (*E*) and strains at break (*ε*) of the three CAB/plasticizer systems measured experimentally. The stress-strain curves of CAB/plasticizer systems were measured firstly and the values of *σ*, *E* and *ε* were calculated from the first linear part of the stress-strain curves (elastic domain), respectively.

The tensile strengths (*σ*) and Young’s moduli (*E*) of CAB/plasticizer systems are in the order of CAB/A3>CAB/Bu-NENA>CAB/GAPA, indicating that the mechanical strength and resistance capacity to deformation of CAB/A3 are best, significantly better than CAB/Bu-NENA and CAB/GAPA. However, the strain at break (*ε*) of CAB/Bu-NENA are higher than CAB/A3 and CAB/GAPA, which means better deformability, but, worst stiffness. Tensile strength test results are consistent with MD simulation results, which further confirm the applicability of MD simulation to the design of energetic materials.

## 5. Conclusions

In the present work, the compatibilities between CAB and seven plasticizers were firstly evaluated numerously by MD simulation and compatible plasticizers are approximately selected. Afterwards, the mechanical properties of CAB with selected plasticizers were calculated by MD simulation. The mixed systems with suitable plasticizers were further characterized experimentally. Simulation results suggest that GAPA, Bu-NENA, A3 and DEGBAA are compatible with CAB well and other plasticizers including PETKAA, TMNTA and EGBAA are incompatible with CAB. The shear moduli (*G*) of optimal CAB/plasticizer systems are similar, but their bulk moduli (*K*), Poisson’s ratios (*μ*), *K/G* ratios and Young’s moduli (*E*) are all in the order of CAB/Bu-NENA >CAB/A3 >CAB/GAPA, suggesting the mechanical properties of CAB/Bu-NENA are the best.

FT-IR characterization suggests that no chemical reaction occurs during the mixing procedure of CAB and plasticizers, and these CAB/plasticizer systems are chemically stable. DSC/DTA analysis further demonstrates that A3, GAPA and Bu-NENA plasticizers are compatible with CAB at level A, and thus these energetic plasticizers are safe for the explosive designs with CAB as binder. It is found by tensile strength measurements that tensile strengths (*σ*) and Young’s moduli (*E*) of CAB/plasticizer systems are in the order of CAB/A3>CAB/Bu-NENA> CAB/GAPA, indicating that the mechanical strength and resistance capacity to deformation of CAB/A3 are best, significantly better than CAB/Bu-NENA and CAB/GAPA. However, the strain at break (*ε*) of CAB/Bu-NENA are higher than CAB/A3 and CAB/GAPA, which means better deformability, but, worst stiffness.

All experimental results are consistent with MD simulation results well, indicating that MD simulation is a suitable study method for energetic material designs. Based on both simulation and experimental results, it can be concluded that A3, Bu-NENA and GAPA are the suitable plasticizers of CAB binder to improve the poor mechanical properties and processing properties of PBXs. Our work has provided a crucial guidance and reference for the formulation of PBXs using CAB as the binder.

## Figures and Tables

**Figure 1 polymers-12-01272-f001:**
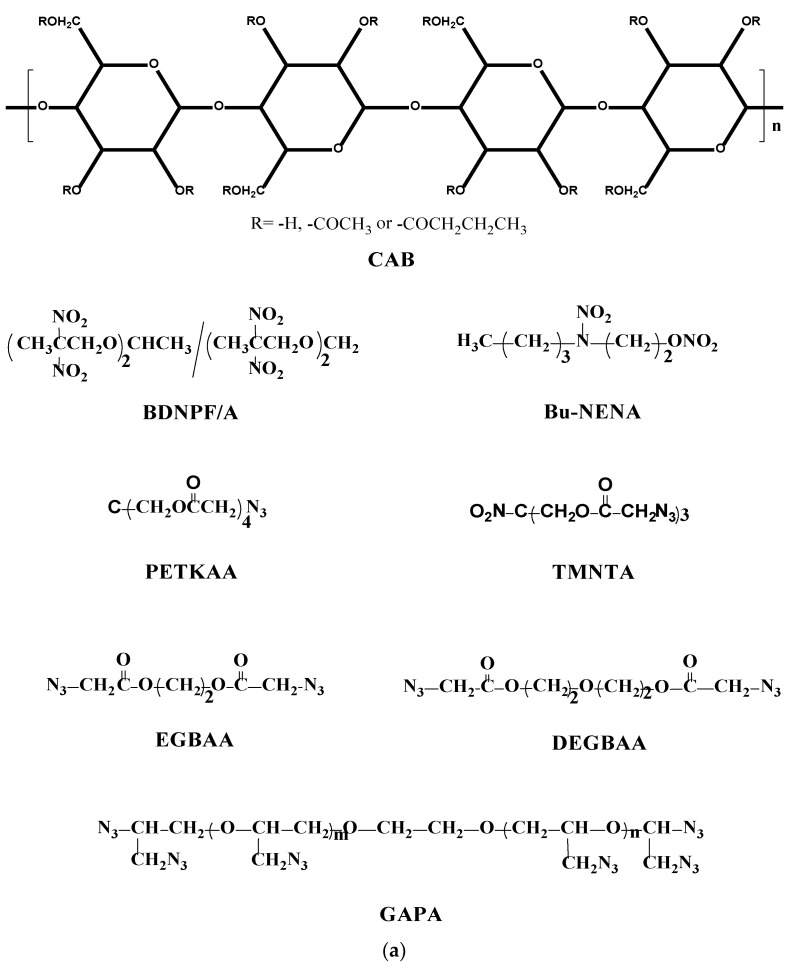
Molecular structures (**a**) and amorphous models (**b**) of CAB/plasticizer systems. CAB = cellulose acetate butyrate.

**Figure 2 polymers-12-01272-f002:**
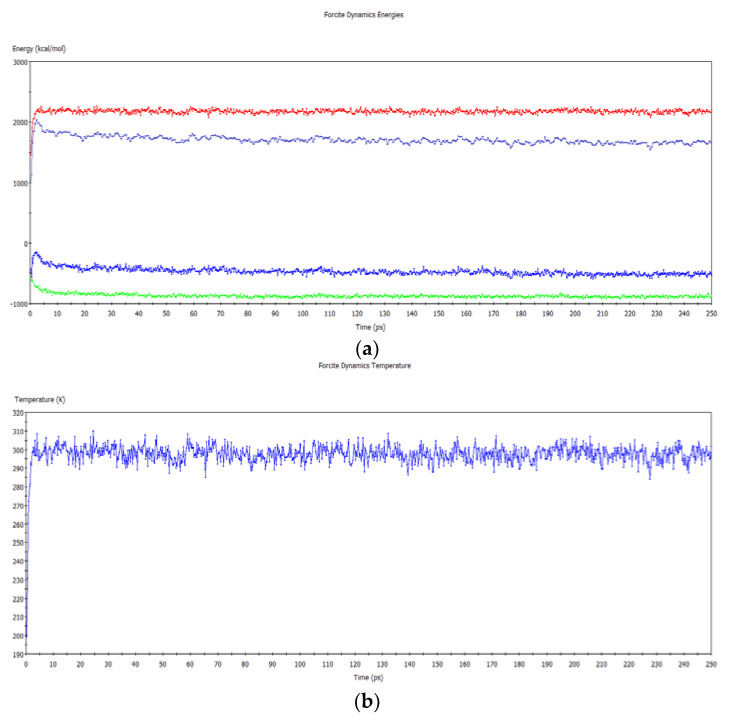
Dynamic energy-time (**a**) and temperature-time (**b**) curves of the molecular dynamics (MD) simulation. The red, light blue, dark blue and green curves in (**a**) represent kinetic energy, total energy, potential energy and non-bonded energy, respectively.

**Figure 3 polymers-12-01272-f003:**
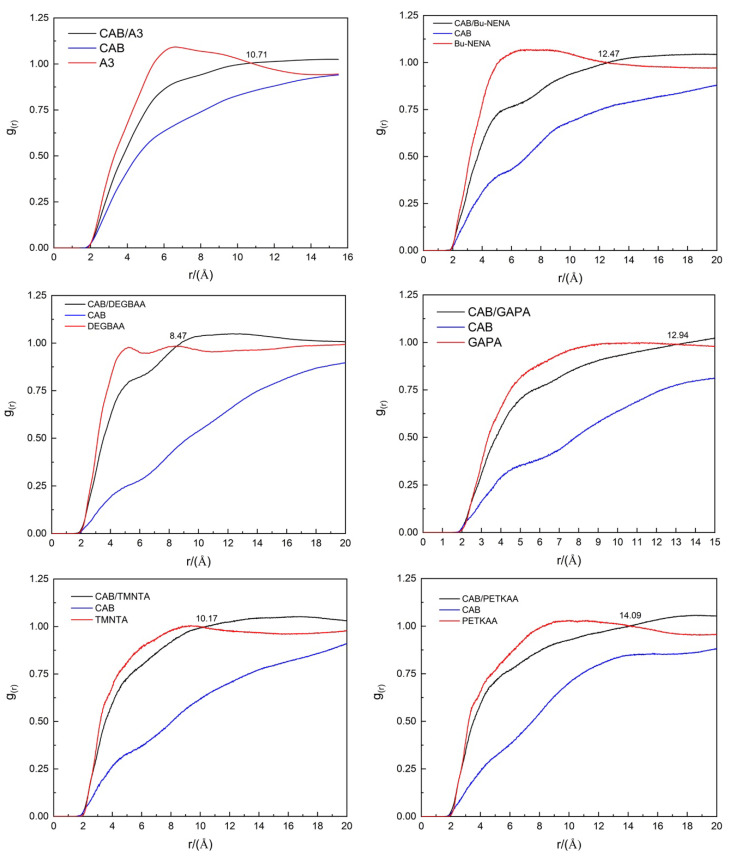
Radial distribution functions (RDF) curves of different CAB/plasticizer systems.

**Figure 4 polymers-12-01272-f004:**
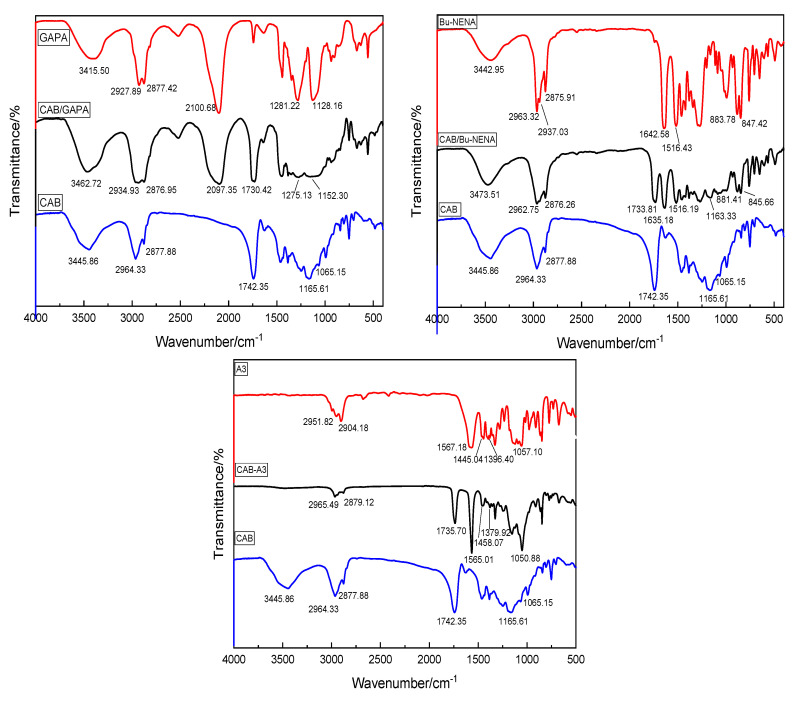
FT-IR spectra of CAB, plasticizer and CAB/plasticizer systems.

**Figure 5 polymers-12-01272-f005:**
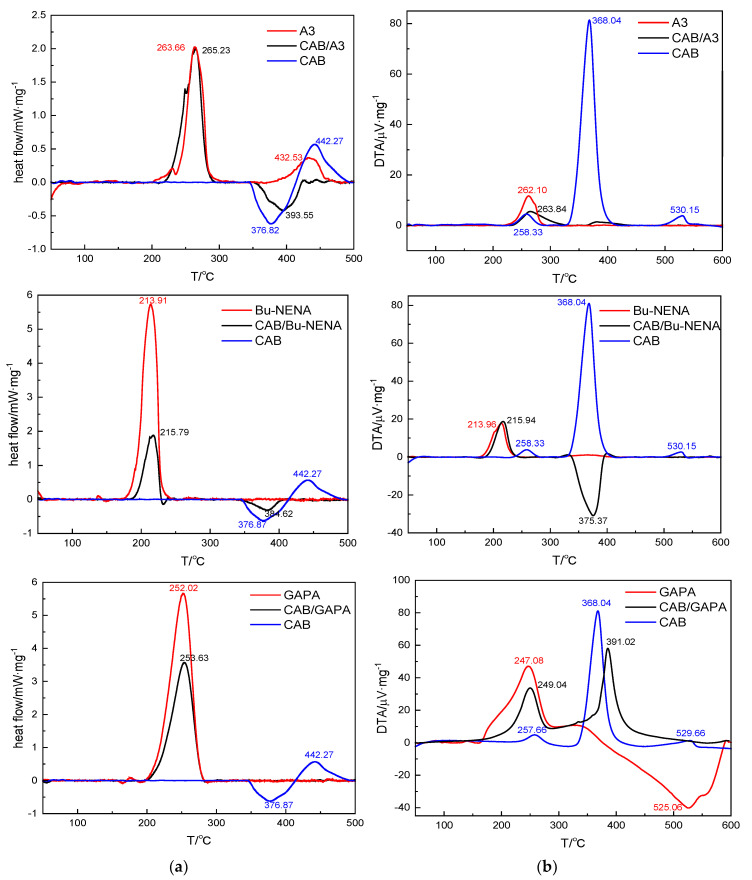
Differential scanning calorimetry (DSC) (**a**) and differential thermogravimetric analysis (DTA) (**b**) curves of CAB, three plasticizers and the CAB/plasticizer systems at the heating rate of 10 K/min.

**Figure 6 polymers-12-01272-f006:**
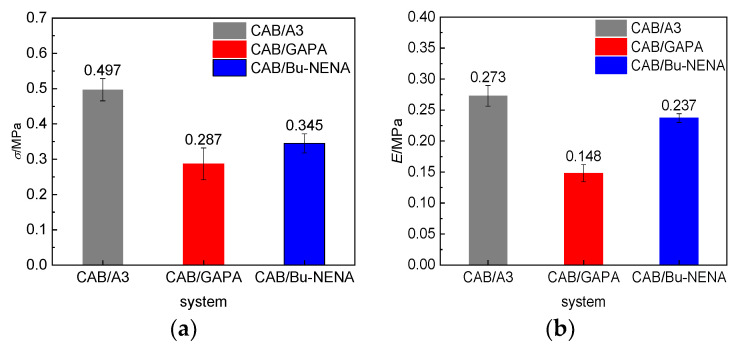
Mechanical properties of CAB/plasticizer systems measured experimentally: (**a**) Tensile strengths (*σ*), (**b**) Young’s moduli (*E*), and (**c**) strains at break *ε*.

**Table 1 polymers-12-01272-t001:** Solubility parameters (*δ_MD_*) of CAB and each plasticizer and the respective differences (|Δ*δ_MD_*|) between the solubility parameters of CAB and different plasticizers.

Component	*δ_MD_*/(J^1/2^·cm^−3/2^)	|Δ*δ_MD_*|/(J^1/2^·cm^−3/2)^
CAB	17.29	0
A3	20.01	2.72
GAPA	20.56	3.27
EGBAA	23.59	6.20
DEGBAA	18.72	1.43
TMNTA	24.55	7.26
Bu-NENA	20.84	3.55
PETKAA	23.30	6.01

**Table 2 polymers-12-01272-t002:** The binding energies (*E_bind_*) and per unit mass binding energies (*E_bind_’*) between CAB and different plasticizers.

System	*E_valence_*/kcal·mol^−1^	*E_vdw_*/kcal·mol^−1^	*E_elect_*/kcal·mol^−1^	*E_bind_*/kcal·mol^−1^	*E_bind_’*/kcal·g^−1^
CAB/A3	0	−448.44	−149.96	598.40	0.71
CAB/GAPA	0	−565.11	−116.36	681.47	0.91
CAB/PETKAA	0	−596.22	−157.32	753.55	1.08
CAB/EGBAA	0	−578.03	−144.21	722.25	2.03
CAB/Bu-NENA	0	−550.03	−115.85	665.88	2.03
CAB/TMNTA	0	−600.74	−153.22	753.96	1.25
CAB/DEGBAA	0	−562.56	−138.11	700.67	1.65

**Table 3 polymers-12-01272-t003:** The mechanical parameters of CAB/plasticizers.

System	*K*/GPa	*G*/GPa	*K/G*	*μ*	*E*/GPa
CAB/A3	2.32	0.86	2.70	0.34	2.29
CAB/GAPA	2.50	0.78	3.20	0.36	2.13
CAB/DEGBAA	1.54	0.91	1.70	0.25	2.27
CAB/Bu-NENA	5.23	0.89	5.87	0.42	2.53
CAB/TMNTA	2.53	1.02	2.49	0.32	2.69
CAB/PETKAA	2.26	1.34	1.68	0.25	3.36
CAB/EGBAA	2.24	0.83	2.69	0.33	2.22

**Table 4 polymers-12-01272-t004:** Δ*T_p_* and Δ*E/E_a_* of different CAB/plasticizer systems and the corresponding compatibility obtained from DSC/DTA.

System	*ΔT_p_/*°C		*(ΔE/E_a_)/%*		Compatibility	
	DSC	DTA	DSC	DTA	DSC	DTA
CAB/A3	1.57	1.74	17.25	19.48	Very good	Very good
CAB/Bu-NENA	1.88	1.98	16.62	11.82	Very good	Very good
CAB/GAPA	1.61	1.96	18.05	18.80	Very good	Very good

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
