# Peer review of "Study on Cellulose Acetate Butyrate/Plasticizer Systems by Molecular Dynamics Simulation and Experimental Characterization"

_polymers, 2020, doi:10.3390/polym12061272_

Round 1
Reviewer 1 Report
My review is in the attached PDF.

Author Response
Dear Editor:
We have revised our manuscript,entitled “Study on cellulose acetate butyrate/plasticizer systems by molecular dynamics simulation and experimental characterization” (manuscript ID polymers-784793) carefully according to the reviewers’ comments. Thanks for the scientific and earnest advices prompted to my manuscript by you and the reviewers. All the revised paragraphs have been highlighted in red color. A list of response to the reviewers’ comments is attached. We have resubmitted the manuscript on your website. We are grateful to you, who made great contribution to improve our paper. If there are any more comments on our paper, please let us know.
Yours sincerely
Yu Chen
Answer to the comments of Reviewer 1:
General comments: This is an interesting and novel study. The introduction and motivation are clear, and the conclusions follow logically from the results. There are several aspects of the simulation methods that should be clarified, as described below. I recommend publication after the authors address these relatively minor comments.
Specific comments
Major
- The authors should state the size (number of atoms, approximate box size) for each system. This is important for interpreting the results of the mechanics simulations. Small systems (<20k atoms) have large fluctuations in the pressure/stress, which can make it challenging to accurately calculate the moduli.
Thanks for the reasonable advice. We have replenished the following explanations in lines 96-99.
The mass ratio of CAB/plasticizer was set to 2:3 and the amounts of CAB molecules (3 chains, 734 atoms) in all amorphous models were the same. The plasticizer contents were approximately in the range of 30~60 molecules (1200~1800 atoms) in their systems, which depended on various kinds of plasticizers.
- A short (250 ps), low-temperature (298 K) simulation is usually not sufficient to equilibrate a glassy polymer. Slow annealing from the rubbery state at high T produces much more accurate results. Rapid cooling often traps the system in a high energy, low density state that doesn’t represent the glass. The authors should briefly comment on the influence of cooling rate/protocol on their results. However, because the MD results are used in a qualitative manner, I do not expect this to affect the overall results and
In our study, a short (250 ps), low-temperature (298 K) simulation is sufficient enough to reach equilibrium and will not greatly influence simulation results.
We have done some anneal simulation of various molecular models, which aims to relax their configurations, lower the potential energy and eliminate the internal stress of polymer chains. The initial temperature was 298K and mid-cycle temperature was set to 500K. The heating ramps per cycle was set to 25, which is a proper step number because too rapid cooling often traps the system in a high energy, low density state that doesn’t represent the glassy state. In the contrast, slow annealing from rubbery state with high Tg can generate more accurate results. After 500 cycles of annealing, the annealed molecular models were obtained for further simulation.
- The authors should provide more detail about the calculation methods for the cohesive energy density (i.e., how was \Delta E calculated?).
We have replenished more statements in lines 108-110 to explain the calculation methods for CED.
In fact, there is a “Cohesive Energy Density” functional option in Forcite module of Materials Studio. After the dynamics simulation are done, we directly apply the “Cohesive Energy Density” functional option to analyze the dynamic trajectory files so that CED and solubility parameter are produced. Therefore, ∆E and other parameters are simulated with use of systematic algorithms and not presented to users.
The specific procedure we add is as followed:
MD simulation of 250 ps was firstly performed in the NVT ensemble at the temperature of 298K to achieve the optimal amorphous models. After the dynamics simulation reaching the equilibrium state, the calculated dynamic trajectory files were analyzed by “Cohesive Energy Density” option of Forcite module so that the cohesive energy density and solubility parameters of CAB and plasticizers were directly obtained.
- The authors only consider solubility in terms of the difference in solubility parameters, \Delta \delta. However, the equation for \Delta H_M also includes the volume fractions of the two components, which might vary significantly depending on the plasticizer. I suggest the authors calculate an estimate of
\Delta H_M using the molecular volumes.
We don’t think volume fractions of plasticizers (Ф2) will influence ∆HM results greatly in our simulation.
There are 3 parts in above equation: VM, Ф1Ф2 and (δ1-δ2)2. In our binary mixing system, CAB and plasticizers are mixed in the mass ratio of 2:3, which means the content of plasticizer is fixed. Compared with CAB, all these plasticizers are micro-molecular compounds and their molecular weight are similar to each other, so Ф2 changes in a small range. The same manner is for VM.
However, solubility parameters of plasticizers (δ2) are much closer to δCAB, so (δCAB-δ2)2 has much more impact on ∆HM.
Apart from that, volume fractions (Ф2) and other parameters are simulated by the systematic algorithms, which means that users don’t need to set these parameters manually.
- The authors should provide more detail about the calculation of the binding energy. E_total is clear, but how were the average energies of the individual molecules calculated? (in the gas phase, or in the mixture?)
Thanks for the reasonable advice of the viewer. We have replenished more explanations in lines 111-114.
Actually we simulated CAB and various plasticizers, respectively, which means we constructed individual system models for CAB and plasticizers separately. So that we can obtain binding energy of CAB and plasticizer, respectively, i.e.: ECAB and Eplasticizer in equation (5).
- The solubility parameter and binding energy appear to show conflicting results. For example, GAPA and A3 are among the most soluble (Table 1), but they have the lowest binding energy (Table 2). Usually the cohesive energy density is based on E_inter, so it’s surprising that the CED and binding energy do not give similar results. I ask that the authors comment on the origin of this apparent discrepancy.
According to our simulation experience with use of Material Studio, the solubility parameter (or CED) calculations results are more accurate criteria for compatibility between polymeric binder and micro-molecular plasticizers.
Binding energy results calculated from MS sometimes seem to be strange or inconsistent with the practical truth. We think this phenomenon probably originates from the imperfection of algorithms on binding energy in MS.
In fact, our experimental results demonstrate A3, GAPA and Bu-NENA are most compatible in our tested 7 plasticizers.
- Please provide more details about how the simulated mechanical tests were conducted. What was the strain rate? What was the maximum strain? Moreover, the high strain rate typically used in simulations explains the large discrepancy with the experiments (~2 GPa vs. ~0.2 MPa). The data are reasonable for MD simulations (e.g., Young’s modulus of a few GPa), but a complete description of the methods should be
We have added more statements about the simulation methods for mechanical tests in lines 118-126.
The simulation mechanical properties tests are performed with “Constant strain” method. The number of steps for each strain was set to 4 and the maximum strain was 3×10-3. The strain rate was used as the automatic systematical setting, which is not presented to users clearly.
Minor
- I suggest that the authors include statistical uncertainty for all the simulated results and provide details about how the uncertainty was calculated. However, I do not consider this critical because the MD results are used in a qualitative manner.
Thanks for the advice of the reviewer. We have replenished more descriptions in corresponding sections, including lines 182-185, 211-214, 237-241 and 286-288.
- Please comment on why the mass ratio of 2:3 CAB:plasticizer was chosen. Is this a typical ratio in PBXs?
Yes. The mass ratio of 2:3 is the most used ratio in fabrication of PBXs and generally it will not be changed. Therefore, we only select 2:3 mass ratio to perform simulation.
- The reference for the COMPASS force field should be corrected. I believe it should be this paper: H. Sun, COMPASS: An ab Initio Force-Field Optimized for Condensed-Phase Applications: Overview with Details on Alkane and Benzene Compounds. The Journal of Physical Chemistry B 102, 7338-7364 (1998).
Thank you! We have replaced the corresponding reference you mentioned to illustrate COMPASS force field better.
- The manuscript says Figure 6c shows the “breaking elongation rate,” but I think it is actually the “strain at break.” Please
Thank you very much for the advice. We have replaced the wrong description with “strain at break” in Fig. 6c. What we actually want to express is the stretching ratio of materials at break and apparently “strain at break” is more professional and accurate.

Reviewer 2 Report
I will be able to comment only on the simulational part of this study; the experimental part of this research is not my field of expertise. The Materials Studio based MD simulations have been performed (in line 63, the “MS simulation” should be changed to Materials Studio simulations, with the proper reference) on a set of some polymer (CAB)-platiciser compositions. The authors try to study the compatibility and mechanics of produced compounds. I have rather strong reservations concerning the performed simulations, and cannot recommend this manuscript for the publication. The performed modelling is done for a very short simulation time, and is carried out at extremely low temperature. If so, the questions arise about the proper equilibration of the systems, which is simply impossible. The details of the simulations are mentioned only very briefly (for example, in line 94 it is said that “ Andersen thermostat was used to achieve the optimal amorphous models” – what does it mean actually? Why the models achieved in this way are optimal, and in what sense?), and the results cannot be reproduced. Some more specific criticism:
- Fig 2: What is shown actually, which energy, what do different colors mean? What do the authors mean by “kinetic temperature”?
- 250ps simulations are extremely short, and no indication is given on the equilibration of the samples. 298K is very low temperature, perhaps below the glass transition, and again, the equilibration of the polymer systmes a tthis temperature is an issue.
- Line 186: “RDF curve of mixed system is higher than each of their own” – this cannot be correct in the whole range of the distances.
- Lines 205-207: the physical reasons on why some plasticizers remain incompativle are not explained. The paper list only some observations, and does not provide any insight.
- Table 3: how were all these characteristics produced? What kind of the mechanical test is performed, at constant strain rate or constant load? How the moduli were calculated from the stress-strain curves? What are the error bars?
I suggest to remove the simulational part completely, or only mention it as some preliminary support to the experimental results (which I cannot judge), and submit the manuscript to a more specialized journal. In any case, the physical reasons and mechanisms for some better compatibility of some plasticizers with CAB should be discussed and explained.
Author Response
Dear Editor:
We have revised our manuscript,entitled “Study on cellulose acetate butyrate/plasticizer systems by molecular dynamics simulation and experimental characterization” (manuscript ID polymers-784793) carefully according to the reviewers’ comments. Thanks for the scientific and earnest advices prompted to my manuscript by you and the reviewers. All the revised paragraphs have been highlighted in red color. A list of response to the reviewers’ comments is attached. We have resubmitted the manuscript on your website. We are grateful to you, who made great contribution to improve our paper. If there are any more comments on our paper, please let us know.
Yours sincerely
Yu Chen
Answer to the comments of Reviewer 2:
- Fig 2: What is shown actually, which energy, what do different colors mean? What do the authors mean by “kinetic temperature”?
Figure 2 shows the dynamics temperature-time and dynamics energy-time curves of the whole simulation process.
Different colors represent different energy curves. We have added the following statements in the caption of Fig.2: “The red, light blue, dark blue and green curves represent kinetic energy, total energy, potential energy and non-bonded energy, respectively, in energy-time figure (a).”
“kinetic temperature” is a writing mistake, whose correct expression should be “dynamics temperature”.
- 250ps simulations are extremely short, and no indication is given on the equilibration of the samples. 298K is very low temperature, perhaps below the glass transition, and again, the equilibration of the polymer systmes at this temperature is an issue.
In our study, a short (250 ps), low-temperature (298 K) simulation is sufficient enough to reach equilibrium and will not greatly influence simulation results.
One of the important factors affecting simulation time is the size of periodic system models. In our periodic box, there are 3 CAB molecule chains (734 atoms, Mn=2630.74g·mol-1) and 30~60 plasticizers molecules (1200~1500 atoms, depending on the kinds of plasticizers), whose size is not so large. And we also have performed some preliminary MD simulations to study the scale of simulation time, whose results demonstrate that the binding energy of systems begin to converge to an equilibrium state after 250 ps, which could be proved by the curves of dynamics temperature-time and energy-time as shown in Fig.2, meaning the time scale of 250ps is a proper boundary.
We have also done proper anneal simulation for CAB and plasticizer molecular models, respectively, to lower potential energy and eliminate irrational internal stress among molecule chains. The initial temperature was 298K and mid-cycle temperature was set to 500K. The heating ramps per cycle was set to 25 and 500 cycles were performed to anneal.
Besides, 298K has very important practical significance for the storage and appliance of our polymer bonded explosives(PBXs) products. Our further products containing CAB and different plasticizers may be stored at 298K for years and that is exactly our motivation and target to conduct such simulation and experimental works, so we have to set the simulation temperature at 298K.
- Line 186: “RDF curve of mixed system is higher than each of their own” – this cannot be correct in the whole range of the distances.
We think it’s possible and we have found some relevant literature to prove our statements.
In fact, RDF represents the possibility of finding a target particle at the distance r away from a given reference particle. In the binary mixed systems, if the intermolecular g(r) is higher than individual component, it means the possibility of finding another kind of particle is larger than finding itself, indicating the interactions between different particles are better those of individual components. Therefore, this can illustrate different particles in the mixed system show the trend of miscibility between each other.
There are other literatures which can prove:
- Macromolecules, 2000, 33(25): 9452-9463. doi:10.1021/ma0011035.
“Miscibility of the melt of iPP and atactic polypropylene(aPP), and immiscibility of the melt of iPP and sPP, could be inferred from the height of the first maximum in the pair correlation functions (intermolecular radial distribution functions).” And there are also some figures showing “RDF curve of mixed system is higher than each of their own” in Figure 9-12 in this paper.
- Physical Review E, 1995, 52(4): 3730. doi: 1103/PhysRevE.52.3730.
“The anion-cation RDF has a first contact maximum at r~2.8 â„«, and a second maximum at r=5.1â„« corresponding to the configuration with a single water molecule between the ions.” Fig.3 shows “RDF curve of mixed system is higher than each of their own” in this paper.
- Journal of Polymer Research, 2017, 24(1): 8. doi: 10.1007/s10965-016-1174-3.
“When heterocontacts between the two components in the blends reach higher g(r) values than the contacts between the same components, miscibility occurs, whereas when this is not the case the system phase separates.” Fig.2 shows “RDF curve of mixed system is higher than each of their own” in this paper.
- Molecular Simulation, 2013, 39(5): 415-422. doi: 10.1080/08927022.2012.738294.
“It has been observed that if a binary system is compatible, the inter-molecular g(r) of AB pair between two different polymers is larger than those of AA and BB pairs.” Fig.3 shows “RDF curve of mixed system is higher than each of their own” in this paper.
- Lines 205-207: the physical reasons on why some plasticizers remain incompativle are not explained. The paper list only some observations, and does not provide any insight.
Thanks for the advice. We have replenished more interpretations in lines 237-241 to explain the physical and chemical structure reason on some plasticizers’ incompatibility with CAB based on your comment.
The nitro and azide groups in their structure may be the original reason of poor physical interactions between these groups and cellulose chains of CAB, which lead to poor compatibility.
- Table 3: how were all these characteristics produced? What kind of the mechanical test is performed, at constant strain rate or constant load? How the moduli were calculated from the stress-strain curves? What are the error bars?
We have added more statements in lines 118-126 to describe the simulation procedure of mechanical properties based on your questions.
These mechanical property parameters are directly calculated from the Forctie module in Materials Studio (version 6.0). To be more specific, we performed NPT and NVT dynamics simulations for system models firstly, and then use “Mechanical Properties” option of Forcite module to conduct the final calculations. Bulk moduli (K), shear moduli (G), Poisson's ratio (μ) of models are directly obtained from “Mechanical Properties” simulation results.
K/G is simple mathematical division calculation and Young's modulus (E) are calculated by the following equation:
The simulation tests are performed with “Constant strain” method. The number of steps for each strain was set to 4 and the maximum strain was 3×10-3. The strain rate was used as the automatic systematical setting, which is not presented to users clearly. But, there are no “stress-strain curves” and no “error bars” produced in Materials Studio.
I suggest to remove the simulational part completely, or only mention it as some preliminary support to the experimental results (which I cannot judge), and submit the manuscript to a more specialized journal. In any case, the physical reasons and mechanisms for some better compatibility of some plasticizers with CAB should be discussed and explained.
Thanks for criticism and advice of the reviewer. The main method of our work is “MD simulation + experimental characterization”. The simulation is the major section to predict compatibility between CAB and plasticizers, while experimental section is the preliminary tool to verify our simulation results. Therefore, we think the MD simulation section should be reserved.

Reviewer 3 Report
First, it is a very nice paper covering both modeling and experiments.
My first remarks is regarding the fact the authors are looking at plasticization, and typically a way to prove plasticization could be to determine Tg and impact of plasticizers on Tg.
- It is also possible to estimate plasticization effect by doing molecular modeling calculation of Tg (quite a lot of literature on this) and to compare with experiments (DSC, TMA).
More details on number of polymer chains, for example in Figure 1.a, it missed what is R in CAB structure.
- What is the degrees of butylation in the structure (the same than experiments ? ie Analytical grade CAB containing 37 wt% butyryl group and 13 wt% acetyl group 108 was provided by Eastman Chemical Company (USA)) and how did they manage the probability of butylation on the different positions?
More details on structural parameters Rg, Ree vs size box (for ex Ree2/Rg2 should be equal to 6).
- Why authors have selected COMPASS force field for CAB? Are there any publications to refer, maybe opls-aa or gromos or charmm are more relevant.
More details regarding equilibrium of the simulation and protocol (for ex: equilibration, NVT, NPT, simulated annealing and final NPT run), I have some doubts that equilibrium could be achieved in 250 ps so if authors could bring more in order to be sure that they analyze a thermodynamically stable structure.
- Is-it possible also to determine structural parameters of polysaccharides such as phi/psi angles and puckering, that s could help to understand if the force field used in robust (see Cellulose 21 (6), 3897-3912, 2014)
There are good descriptions of solubility parameters, binding energy and rdf calculations allowing to rank the different plasticizers.
- Is-it possible for the authors to compare calculated solubility parameters of CAB with some experimental data that could help to support their force field?
- The compatibility between a polymer and a different compound has been obtained for one ratio polymer/plasticizer. Are they referring to enthalpy of mixing? Because in this case it should interesting to test other ratio and plot Enthalpy of mixing as function of plasticizer content 8more relevant to determine the compatibility between two components).
- It is well known that the distance ranges of hydrogen bond and van der Waals force are 0-3.10 Å and 3.10-5.00 Å , respectively line 197: can the authors provide a reference?
More details regarding the simulation of mechanical properties.
- Until which deformation modulus are calculated (for example people are using anisotropic Barostat).
- The materials with the Poisson's ratios in the range between 0.2 and 0.4 are generally considered to have good plasticity line 236: can the authors provide a reference?
Author Response
Dear Editor:
We have revised our manuscript,entitled “Study on cellulose acetate butyrate/plasticizer systems by molecular dynamics simulation and experimental characterization” (manuscript ID polymers-784793) carefully according to the reviewers’ comments. Thanks for the scientific and earnest advices prompted to my manuscript by you and the reviewers. All the revised paragraphs have been highlighted in red color. A list of response to the reviewers’ comments is attached. We have resubmitted the manuscript on your website. We are grateful to you, who made great contribution to improve our paper. If there are any more comments on our paper, please let us know.
Yours sincerely
Yu Chen
Answer to the comments of Reviewer 3:
- More details on number of polymer chains, for example in Figure 1.a, it missed what is R in CAB structure.
Thanks for advice of the reviewer. We indeed forgot to illustrate what “R” means in the structure of CAB. Actually, “R” means different substitute groups, i.e.: R= -H, -COCH3 or –COCH2CH2CH3
We have also replenished this claim in Figure 1.a.
- What is the degrees of butylation in the structure (the same than experiments ? ie Analytical grade CAB containing 37 wt% butyryl group and 13 wt% acetyl group 108 was provided by Eastman Chemical Company (USA)) and how did they manage the probability of butylation on the different positions?
The degrees of butylation in simulation are the same as the experiments.
In our experimental section, we utilize CAB with 37 wt% butyryl group and 13 wt% acetyl group; therefore, we also constructed CAB model with the same functional group contents manually in simulation. As the numbers of butyryl and acetyl groups can only be integer, the group contents are approximate and not so restricted to 37wt% and 13wt%.
The butylation positions were controlled manually when we constructed CAB model and are randomly located on the CAB chain.
- More details on structural parameters Rg, Ree vs size box (for ex Ree2/Rg2 should be equal to 6).
Thanks for the advice and we have added more explanations in lines 96-99 to illustrate structure information of system models.
The periodic model boxs are constructed with use of “Amourphous Cell” module in Materials Studio, and the sizes of box are a=25.7, b=25.7, c=25.7 (accurate sizes depend on different kinds of CAB/plasticizer). The boxes contain 3 CAB chains (734 atoms, Mn=2630.74g·mol-1) and 30~60 plasticizers molecules (1200~1500 atoms, depending on the kinds of plasticizers).
As for Rg, Ree and other parameters, we just follow the automatic settings of Materials Studio software. There are some other literatures about CAB or similar cellulose derivatives with the automatic settings of relevant parameters, e.g.:
- A molecular dynamics study and detonation parameters calculation of 5, 5’-dinitramino-3, 3’-bi [1, 2, 4-triazolate] carbohydrazide salt (CBNT) and its PBXs[J]. Journal of Energetic Materials, 2019: 1-12. doi: 1080/07370652.2019.1684595.
- Effect of toughener on desensitizer and 2, 4, 6, 8, 10, 12-hexanitro-2, 4, 6, 8, 10, 12-hexaazaisowurtzitane (HNIW) based polymer bonded explosives (PBXs)[J]. Materials Express, 2017, 7(6): 529-535. doi: 1166/mex.2017.1399.
- Investigation of the effect of the CAB/A3 system on HNIW-based PBXs using molecular dynamics[J]. Journal of molecular modeling, 2018, 24(7): 186. doi: 1007/s00894-018-3670-3.
- Oligo-and Polyfluorenes Meet Cellulose Alkyl Esters: Retention, Inversion, and Racemization of Circularly Polarized Luminescence (CPL) and Circular Dichroism (CD) via Intermolecular C–H/Oî—» C Interactions[J]. Macromolecules, 2017, 50(5): 1778-1789. doi: 1021/acs.macromol.6b02762.
- Why authors have selected COMPASS force field for CAB? Are there any publications to refer, maybe opls-aa or gromos or charmm are more relevant.
COMPASS force field is widely applicable for various polymers, micromolecules and ions. This force field is widely applied for simulation of cellulose, CAB or similar cellulose derivatives by researchers, e.g.:
- Selection of optimal polymerization degree and force field in the molecular dynamics simulation of insulating paper cellulose[J]. Energies, 2017, 10(9): 1377. doi: 3390/en10091377.
- Force field properties in molecular simulation of amorphous region in cellulose insulation paper[J]. Oxidation Communications, 2016, 39(1 A): 1236-1246.
- Evaluation of reactive force fields for prediction of the thermo-mechanical properties of cellulose Iβ[J]. Computational Materials Science, 2015, 109: 330-340. doi: 1016/j.commatsci.2015.06.040.
- Molecular modelling of cellulose dissolution[J]. Journal of Computational and Theoretical Nanoscience, 2013, 10(11): 2639-2646. doi: 1166/jctn.2013.3263.
- Validating empirical force fields for molecular-level simulation of cellulose dissolution[J]. Computational and Theoretical Chemistry, 2012, 984: 119-127. doi: 1016/j.comptc.2012.01.020.
- Modelling the crystalline deformation of native and regenerated cellulose[J]. Cellulose, 2006, 13(3): 291-307. doi: 1007/s10570-006-9046-3.
- A molecular dynamics study and detonation parameters calculation of 5, 5’-dinitramino-3, 3’-bi [1, 2, 4-triazolate] carbohydrazide salt (CBNT) and its PBXs[J]. Journal of Energetic Materials, 2019: 1-12. doi: 1080/07370652.2019.1684595.
- Effect of toughener on desensitizer and 2, 4, 6, 8, 10, 12-hexanitro-2, 4, 6, 8, 10, 12-hexaazaisowurtzitane (HNIW) based polymer bonded explosives (PBXs)[J]. Materials Express, 2017, 7(6): 529-535. doi: 1166/mex.2017.1399.
- Investigation of the effect of the CAB/A3 system on HNIW-based PBXs using molecular dynamics[J]. Journal of molecular modeling, 2018, 24(7): 186. doi: 1007/s00894-018-3670-3.
- More details regarding equilibrium of the simulation and protocol (for ex: equilibration, NVT, NPT, simulated annealing and final NPT run), I have some doubts that equilibrium could be achieved in 250 ps so if authors could bring more in order to be sure that they analyze a thermodynamically stable structure.
In our study, the time scale of 250 ps is enough for systems to reach equilibrium, and it will not greatly influence simulation results.
One of the important factors affecting simulation time is the size of periodic system models. In our periodic box, there are 3 CAB molecules (734 atoms, Mn=2630.74g·mol-1) and 30~60 plasticizers molecules (1200~1500 atoms, depending on the kinds of plasticizers), whose size is not so large. And we also have performed some preliminary MD simulations to study the scale of simulation time, whose results demonstrate that the binding energy of systems begin to converge to an equilibrium state after 250 ps, meaning the time scale of 250ps is a proper boundary.
- Is-it possible also to determine structural parameters of polysaccharides such as phi/psi angles and puckering, that s could help to understand if the force field used in robust (see Cellulose 21 (6), 3897-3912, 2014)
There are good descriptions of solubility parameters, binding energy and rdf calculations allowing to rank the different plasticizers.
The whole simulations are performed in Materials Studio and the study focus on the intermolecular interactions between plasticizer molecules and CAB polymer chains. These structural parameters like phi/psi angles and puckering are more specified to forces among atoms or chain segments, which are not the advantages of MS. Maybe we can utilize Gaussian, MAPS or other simulation programs to focus on structural parameters of polysaccharides in the future.
- Is-it possible for the authors to compare calculated solubility parameters of CAB with some experimental data that could help to support their force field?
Thanks for the advice of the reviewer. We have quoted some relevant literature about solubility parameters of CAB and compare them with our calculated solubility parameters in lines 182-184.
The experimental solubility parameter of CAB is 18.87±2.47 or 18.0( J1/2·cm-3/2), which is similar to our simulational result of 17.29( J1/2·cm-3/2).
The reference literatures are:
- Determination of Hansen solubility parameters for the solid surface of cellulose acetate butyrate by inverse gas chromatography[J]. Journal of Macromolecular Science, Part B, 2011, 50(3): 551-562. doi: 10.1080/00222341003784527.
- FTIR imaging coupled with multivariate analysis for study of initial diffusion of different solvents in cellulose acetate butyrate films[J]. Cellulose, 2008, 15(1): 23-33. doi: 10.1007/s10570-007-9173-5.
- The compatibility between a polymer and a different compound has been obtained for one ratio polymer/plasticizer. Are they referring to enthalpy of mixing? Because in this case it should interesting to test other ratio and plot Enthalpy of mixing as function of plasticizer content 8more relevant to determine the compatibility between two components).
No, the ratio refer to mass ratio of CAB/plasticizer, which is set to 2:3. Because this ratio is mostly used in further industry production formula, and generally it’s not changed. Therefore, we only select 2:3 mass ratio and ignore other ones.
- It is well known that the distance ranges of hydrogen bond and van der Waals force are 0-3.10 Å and 3.10-5.00 Å , respectively line 197: can the authors provide a reference?
Thanks for the advice of the reviewer. We have quoted some relevant literature to prove our statements. The following references give elaborate information about distance ranges of hydrogen bond and van der Waals force, and we have also replenished these references in our paper.
- Crystal morphology of 3, 4-bis (3-nitrofurazan-4-yl) furoxan (DNTF) in a solvent system: molecular dynamics simulation and sensitivity study[J]. CrystEngComm, 2016, 18(16): 2843-2851. doi: 10.1039/C6CE00049E.
- Crystal morphology of dihydroxylammonium 5, 5′-bistetrazole-1, 1′-diolate (TKX-50) under solvents system with different polarity using molecular dynamics[J]. Computational Materials Science, 2019, 168: 48-57. doi: 10.1016/j.commatsci.2019.05.060.
- Interactions and physical properties of energetic poly-(phthalazinone ether sulfone ketones)(PPESKs) and ε-hexanitrohexaazaisowurtzitane (ε-CL-20) based polymer bonded explosives: a molecular dynamics simulations[J]. Structural Chemistry, 2019, 30(3): 1041-1055. doi: 1007/s11224-018-1225-y.
- Morphology control of 3-nitro-1, 2, 4-triazole-5-one (NTO) by molecular dynamics simulation[J]. CrystEngComm, 2018, 20(40): 6252-6260. doi: 1039/C8CE00756J.
- Until which deformation modulus are calculated (for example people are using anisotropic Barostat).
Until uniaxial deformation.
Young’s Modulus defines the relationship between stress (force per unit area) and strain (proportional deformation) in a material in the linear elasticity regime of a uniaxial deformation.
- The materials with the Poisson's ratios in the range between 0.2 and 0.4 are generally considered to have good plasticity line 236: can the authors provide a reference?
Thanks for the advice of the reviewer. We have quoted some relevant literature to prove our statements. The following references give elaborate information about Poisson's ratios, and we have also replenished these references in our paper.
- Miscibility, glass transition temperature and mechanical properties of NC/DBP binary systems by molecular dynamics[J]. Propellants, Explosives, Pyrotechnics, 2018, 43(6): 559-567. doi: 1002/prep.201700290.
- Molecular dynamics simulations on miscibility, glass transition temperature and mechanical properties of PMMA/DBP binary system[J]. Journal of Molecular Graphics and Modelling, 2018, 84: 182-188. doi: 10.1016/j.jmgm.2018.07.005.
- Interactions between poly-(phthalazinone ether sulfone ketone)(PPESK) and TNT or TATB in polymer bonded explosives: a molecular dynamic simulation study[J]. Journal of molecular modeling, 2017, 23(12): 334. doi: 10.1007/s00894-017-3492-8.
- Theoretical simulations on the glass transition temperatures and mechanical properties of modified glycidyl azide polymer[J]. Computational Materials Science, 2017, 139: 132-139. doi: 10.1016/j.commatsci.2017.07.022.

Round 2
Reviewer 2 Report
I have read the authors response and the updated manuscript. I see that the authors took into account the criticism, and the manuscript is better now.
Still, I advise the authors to carefully check the English, I think it can be improved further. One other small point: the glass transition temperature of the investigated compounds should be mentioned. I expect 250K is below Tg. In this case it is simply impossible to produce equilibrated samples, and 250ps is very short...and it would not be better if it would be 250 years…. Practically, of course, energy can be considered as (almost) constant, but it, nevertheless, changes, this is called ageing. This can be also added to the manuscript.
I think the manuscript can be published, but, again, English should be improved.
Author Response
Dear Editor:
We have revised our manuscript,entitled “Study on cellulose acetate butyrate/plasticizer systems by molecular dynamics simulation and experimental characterization” (manuscript ID polymers-784793) carefully according to the reviewers’ comments. Thanks for the scientific and earnest advices prompted to my manuscript by you and the reviewers. All the revised paragraphs have been highlighted in red color. A list of response to the reviewers’ comments is attached. We have resubmitted the manuscript on your website. We are grateful to you, who made great contribution to improve our paper. If there are any more comments on our paper, please let us know.
Yours sincerely
Yu Chen
Answer to the comments of Reviewer 2:
I advise the authors to carefully check the English, I think it can be improved further. One other small point: the glass transition temperature of the investigated compounds should be mentioned. I expect 250K is below Tg. In this case it is simply impossible to produce equilibrated samples, and 250ps is very short...and it would not be better if it would be 250 years…. Practically, of course, energy can be considered as (almost) constant, but it, nevertheless, changes, this is called ageing. This can be also added to the manuscript. I think the manuscript can be published, but, again, English should be improved.
Response: Thanks for the accurate advice and criticism of the reviewer. We have asked help from professional institution to improve the language and polished our English writing in the paper. We have also supplied relevant information about glass transition temperature (Tg) of CAB (403K) and plasticizers (190K~240K) in corresponding parts (lines 86-89). Besides, we have added the thoughts and discussion about simulation energies in lines 110-112.

Reviewer 3 Report
Dear authors,
Thanks for reviewing the manuscript which seems well improved.
It is still missing a very important information linked with plasticization:
See my first remarks:
My first remarks is regarding the fact the authors are looking at plasticization, and typically a way to prove plasticization could be to determine Tg and impact of plasticizers on Tg.
- It is also possible to estimate plasticization effect by doing molecular modeling calculation of Tg (quite a lot of literature on this) and to compare with experiments (DSC, TMA).
With my best regards.
Author Response
Dear Editor:
We have revised our manuscript,entitled “Study on cellulose acetate butyrate/plasticizer systems by molecular dynamics simulation and experimental characterization” (manuscript ID polymers-784793) carefully according to the reviewers’ comments. Thanks for the scientific and earnest advices prompted to my manuscript by you and the reviewers. All the revised paragraphs have been highlighted in red color. A list of response to the reviewers’ comments is attached. We have resubmitted the manuscript on your website. We are grateful to you, who made great contribution to improve our paper. If there are any more comments on our paper, please let us know.
Yours sincerely
Yu Chen
Answer to the comments of Reviewer 3:
It is still missing a very important information linked with plasticization: See my first remarks: My first remarks is regarding the fact the authors are looking at plasticization, and typically a way to prove plasticization could be to determine Tg and impact of plasticizers on Tg. It is also possible to estimate plasticization effect by doing molecular modeling calculation of Tg (quite a lot of literature on this) and to compare with experiments (DSC, TMA).
Response: Thanks for the remarkable advice of the viewer. To be honest, the main target of our work is to study compatibilities between CAB binder and various plasticizers and select proper CAB/plasticizer systems which can meet the demand of compatibility and mechanical properties for further application in polymer bonded explosives (PBXs). In other words, CAB and plasticizers will make different contributions in our further PBXs formula. But they come into consideration of application under the premise of good compatibility. Therefore, compatibility and mechanical properties are most important research objectives in current work. We will accept the viewer’s criticism and take Tg of CAB/plasticizer into account for further investigation.
